# ARE WE IN (A)SYNC?:
# GUIDANCE FOR EFFICIENT FEDERATED LEARNING

## ABSTRACT

Federated Learning (FL) methods have widely adopted synchronous FL (syncFL), where a server distributes and aggregates the model weights with clients in coordinated rounds. As syncFL suffers from low resource utilization on clients with heterogeneous computing power, asynchronous FL (asyncFL), which allows the server to exchange models with available clients continuously, has been proposed. Despite numerous studies on syncFL and asyncFL, how they differ in training time and resource efficiency is still unclear. Given the training and communication speed of participating clients, we present a formulation of time and resource usage on syncFL and asyncFL. Our formulation weights asyncFL against its inefficiencies stemming from stale model updates, enabling more accurate comparison to syncFL in achieving the same objectives. Unlike previous findings, the formulation reveals that no single approach always works better than the other regarding time and resource usage. Our experiments across five datasets show that the formulation predicts relative time and resource usage of syncFL and asyncFL with up to $5.5\times$ smaller root-mean-square error (RMSE) compared to the baseline methods. We envision our formulation to guide FL practitioners in making informed decisions between syncFL and asyncFL, depending on their resource constraints.

## 1 INTRODUCTION

Federated Learning (FL) has emerged as a machine learning paradigm that performs decentralized model training across multiple devices using their locally stored data (McMahan et al., 2017). Most prior FL approaches adopted synchronous FL (syncFL), in which a central server distributes and aggregates model weights with client devices in coordinated rounds (Li et al., 2020; Karimireddy et al., 2020; Reddi et al., 2021; Acar et al., 2021). While syncFL ensures consistency in model training, it often leads to suboptimal resource utilization, as clients who finish training early wait for other slower clients to complete (Chen et al., 2020; Xu et al., 2021). To address this issue, asynchronous FL (asyncFL) has been proposed (Xie et al., 2019; Chai et al., 2021; Wu et al., 2021; Nguyen et al., 2022). In asyncFL, the central server continuously exchanges models with available clients, aiming to harness the full potential of each client's computational power.

We aim to answer the following question: *how do syncFL and asyncFL differ in training time and resource efficiency?* Despite extensive research on syncFL and asyncFL, a definitive understanding of their relative efficiency in time and resource usage remains unclear. Prior studies (Nguyen et al., 2022; Huba et al., 2022) have empirically studied the relative efficiency of asyncFL compared to syncFL. However, these findings are not easily generalizable as the investigations were confined to specific setups (e.g., parameters) of syncFL and asyncFL. Moreover, these conclusions conflict with others (Wu et al., 2021; Sun et al., 2023), indicating a lack of clarity on variables impacting the efficiency of both methods. Such an ambiguity makes it challenging for FL practitioners with limited resources to decide which of the two methods to choose.

To address this problem, we introduce a formulation that quantifies the time and resource usage of syncFL and asyncFL based on the given training and communication time of participating clients. Our formulation accounts for the inefficiencies in asyncFL that arise from stale model updates, providing more accurate comparison with syncFL. Our analysis based on the formulation reveals *neither syncFL nor asyncFL universally outperforms the other in terms of time and resource usage*, which contradicts previous findings (Xie et al., 2019; Nguyen et al., 2022; Huba et al., 2022).

We found that their efficiency is influenced by the distribution and scale of the client training and communication times, along with the choice of parameters such as concurrency.

We evaluated the accuracy of our formulation on five datasets involving up to 21,876 FL clients, comparing the formulated time and resource usage values with the actual results from syncFL and asyncFL runs. The results demonstrate that the formulation predicts the relative efficiency of syncFL and asyncFL with up to $5.5\times$ less root-mean-square error (RMSE) compared to the baseline approaches. We further investigate the applicability of our formulation on different FL optimizations (McMahan et al., 2017; Li et al., 2020; Reddi et al., 2021). Our findings suggest that our formulation could support FL practitioners to better understand the efficiency of syncFL and asyncFL before FL execution, guiding their choices on which approaches and parameters to operate.

## 2 BACKGROUND

### 2.1 SYNCHRONOUS AND ASYNCHRONOUS FEDERATED LEARNING

For syncFL and asyncFL, we refer to the most commonly used algorithms, *FedAvg* (McMahan et al., 2017) and *FedBuff* (Nguyen et al., 2022), respectively. We explain each as follows:

**Synchronous Federated Learning.** Suppose we have $n$ FL clients and a server. FedAvg progresses on a *round* basis, where the server initiates a round by randomly sampling $k$ clients ($k \leq n$). Let $d_i$ denote the number of training samples for a sampled client $i$ where $1 \leq i \leq k$. The total sum of training samples is expressed as $m = \sum_{i=1}^{k} d_i$. The server distributes model weights $w^j$ for round $j$ to the sampled clients. Each client $i$ individually trains a model and responds with the updated weights $w_i^j$. The server waits for clients to finish training and synchronously aggregates the model updates to a new global model as $w^{j+1} \leftarrow \sum_{i=1}^{k} \frac{d_i}{m} w_i^j$.

**Asynchronous Federated Learning.** FedBuff uses two key parameters: (1) Concurrency $c$ and (2) aggregation goal $k$. FedBuff maintains $c$ number of clients training concurrently during FL. At an FL system with $n$ clients, a server starts training with $c$ randomly sampled clients. The server stores the client update in a buffer whenever a client finishes training. Then, the server randomly selects a non-training client for training, maintaining $c$ training clients. When the buffer reaches size $k$, the server aggregates the buffered updates into a new model. Note that in asyncFL, the server model could be updated while clients train, which could let some clients end up training on a stale global model. Such staleness is expressed in $\tau_i$, a version difference between the current global model and the global model trained by client $i$. FedBuff performs aggregation as in FedAvg, except that client updates are each multiplied by $\frac{1}{\sqrt{1+\tau_i}}$, similar to Xie et al. (2019).

### 2.2 FL EFFICIENCY METRICS

We primarily focus on two key metrics that characterize the efficiency of FL execution: (1) *time* and (2) *resource usage*. *Time* denotes the wall-clock time required for a training model to achieve a target test accuracy (Lai et al., 2021). *Resource usage* represents the sum of resources clients use for on-device training and model communication to reach the target test accuracy. As in Abdelmoniem et al. (2023), we quantify resource usage as the time clients spend for training and communication, which is proportional to various resource types such as energy consumption. These metrics collaboratively demonstrate how effectively the FL system could utilize the resources to accelerate FL execution.

We formulate the above metrics for syncFL and asyncFL to guide FL deployers in choosing the most suitable approach before FL execution (discussed in Section 3). For example, if the FL deployers are constrained by time for FL executions, they could leverage our formulation to evaluate the metrics and select an option that optimizes time within their resource budget.

### 2.3 RELATED WORK

**Federated Learning Approaches.** McMahan et al. (2017) introduced the concepts of FL specifically in the form of syncFL, by proposing the most commonly used FL algorithm, FedAvg. Most

subsequent optimizations in the FL domain were constructed on syncFL (Li et al., 2020; Karimireddy et al., 2020; Reddi et al., 2021; Acar et al., 2021). However, syncFL faces challenges with low resource utilization efficiency on heterogeneous clients (Chen et al., 2020; Xu et al., 2021). This is because faster-training clients are often left idle in syncFL, waiting for other clients to complete. To mitigate such a problem, researchers proposed various solutions, such as: configuring a deadline for each round (Nishio & Yonetani, 2019), allowing slower clients to train on a sub-model (Diao et al., 2021; Horváth et al., 2021) or train fewer samples (Shin et al., 2022), or oversampling the clients at a round and partially accepting the faster updates (Li et al., 2019; Lai et al., 2021). However, these methods often led to the waste of client resources or suboptimal model accuracy due to the partial exclusion of clients or not fully adhering to the training process (Abdelmoniem et al., 2022; 2023). To harness the full capabilities of computational resources on FL clients, asyncFL has been introduced (Wu et al., 2021; Nguyen et al., 2022; Chai et al., 2021; Sun et al., 2023), where the server continuously exchanges model weights with available clients. Since asyncFL can lead to stale model updates that might be noisier compared to syncFL, solutions such as weight decaying (Park et al., 2021) or dropout regularization (Dun et al., 2023) have been proposed. Moreover, the convergence of asyncFL methods has been theoretically validated (Fraboni et al., 2023).

**Efficiency Comparison on Synchronous and Asynchronous Federated Learning.** A number of previous studies discussed the efficiency difference between syncFL and asyncFL: Wu et al. (2021) and Sun et al. (2023) highlighted that the communication cost of asyncFL is much higher compared to syncFL. Xie et al. (2019) and Zhang et al. (2023) featured that asyncFL achieves faster convergence than syncFL. Others focused on empirically investigating a difference of FL efficiency between syncFL and asyncFL. Nguyen et al. (2022)'s experimental results on three datasets (CelebA, Sent140, and CIFAR-10) show that syncFL requires about $3.3\times$ more client updates compared to asyncFL to reach the target accuracy. Huba et al. (2022) deployed syncFL and asyncFL to train a language model on millions of mobile users, showing asyncFL achieves $5\times$ less time in achieving accuracy goal with $8\times$ less client updates compared to syncFL. These empirical findings are hardly generalizable as they were under specific settings (e.g., 1,000 concurrency and 10 aggregation goal). Moreover, the fact that the prior findings (e.g., Wu et al. (2021) and Sun et al. (2023) vs. Nguyen et al. (2022) and Huba et al. (2022)) are in disagreement suggests that there still exists a gap in understanding the factors which influence the performance of the two approaches. Some prior studies (Koloskova et al., 2022; Mishchenko et al., 2022) offer theoretical convergence rates for asynchronous SGD based on maximum and average staleness. However, these findings are not directly applicable to the widely used buffered asyncFL method and overlook resource efficiency.

In summary, it still remains unclear how syncFL and asyncFL differ in time and resource efficiency. We present a formulation that quantifies the time and resource usage of both approaches. We further conduct an analysis on our formulation and highlight that the training and communication speeds of clients influence the efficiency of syncFL and asyncFL. We also demonstrate how the efficiencies vary based on different concurrency parameters.

## 3 THEORETICAL FORMULATION

We formulate the time and resource usage required for syncFL and asyncFL to reach the target accuracy. Our formulation assumes that the goal is achieved after $p$ global model updates. Thus, for syncFL, we formulate with an objective to complete $p$ rounds. For asyncFL, we down-weight each global model update based on its staleness (Nguyen et al., 2022) (e.g., a model update could be measured as 0.8 model update due to staleness), and formulate the time and resource usage to reach $p$ model updates, as detailed in Section 3.2.

Let $N = \{1, 2, \ldots, n\}$ denote a set of $n$ FL clients' indices, from client 1 to client $n$. Assuming clients train and communicate models on different hardware setups, we define a set $T = \{t_1, t_2, \ldots, t_n\}$, where $t_i$ indicates a time that client $i$ ($i \in N$) takes to complete the following: (1) download the global model weights, (2) train the model, and (3) upload the local model weights. Without loss of generality, let $t_1 \le t_2 \le \cdots \le t_n$. For asyncFL, $k$ denotes the aggregation goal, i.e., a number of client updates required to update the global model; thus, $k$ is equivalent to the number of clients sampled per round in syncFL. $c$ indicates the concurrency parameter of asyncFL.

### 3.1 SYNCHRONOUS FEDERATED LEARNING

**Time.** As in FedAvg (McMahan et al., 2017), $k$ clients are randomly sampled among total $n$ clients at each round of syncFL. A round duration is determined by the slowest sampled client, which is one of the following: $\{t_k, t_{k+1}, \ldots, t_n\}$. Among $\binom{n}{k}$ of client selection cases, the number of cases which $t_i$ ($k \leq i \leq n$) becomes the round duration is $\binom{i-1}{k-1}$, i.e., number of cases of choosing $k-1$ clients among $\{t_1, t_2, \ldots, t_{i-1}\}$ after choosing $t_i$. Thus, the expected per-round time of syncFL is:

$$\frac{t_k \binom{k-1}{k-1} + t_{k+1} \binom{k}{k-1} + \cdots + t_n \binom{n-1}{k-1}}{\binom{n}{k}}. \tag{1}$$

Thus, the averge time that elapses for syncFL to complete $p$ rounds is $p \times$ Eq.(1).

**Resource Usage.** Average resource usage from a sampled client at a round is $\overline{T}$. Thus, the resource usage required for $p$ rounds with syncFL is $pk\overline{T}$.

### 3.2 ASYNCHRONOUS FEDERATED LEARNING

**Time.** Formulating the time of asyncFL execution involves two tasks: (1) Calculating the number of global model updates until time $x$ with given aggregation goal $k$ and concurrency $c$, and (2) measuring the staleness involved with each global model updates. Based on these two information, we could find time $x$ which asyncFL system achieves $p$ global model updates after accounting staleness. To this end, we define a function $f(T, c, i)$ that denotes an expected portion of time in which client $i$ participated in FL (i.e., model training and communication) at asyncFL setting, where $c \leq |T|$ indicates concurrency and $0 \leq f(T, c, i) \leq 1$ for $i \in N$. For example, if the client $i$ participated in asyncFL for $t_i$ time while the whole training process takes time $t_{whole}$ to reach the target accuracy, then $f(T, c, i) = \frac{t_i}{t_{whole}}$.

**Lemma 1.** *Let $e_c(T)$ denote an elementary symmetric polynomial defined as:* $e_c(T) = \sum_{i_1 < i_2 < \cdots < i_c} t_{i_1} t_{i_2} \ldots t_{i_c}$, *for* $t_{i_1}, t_{i_2}, \ldots, t_{i_c} \in T$. *Then:*

$$f(T, c, i) = \frac{t_i e_{c-1}(T \setminus \{t_i\})}{e_c(T)}.$$

*Proof.* Refer to Appendix A for the proof. $\square$

Given the client times set $T$, we could measure the portion of time which each client participated. This allows us to further measure the number of model updates given by each client, as follows:

**Corollary 1.** *Let $g(T, c, i, x)$ be an expected number of updates from client $i$ until arbitrary time $x$:*

$$g(T, c, i, x) = \frac{x f(T, c, i)}{t_i} = \frac{x e_{c-1}(T \setminus \{t_i\})}{e_c(T)}. \tag{2}$$

*Since $\sum_{i \in N} e_c(T \setminus \{t_i\}) = (n - c)e_c(T)$, an expected portion of updates from client $i$ among all updates given by total $n$ clients:*

$$\frac{g(T, c, i, x)}{\sum_{j \in N} g(T, c, j, x)} = \frac{\frac{x e_{c-1}(T \setminus \{t_i\})}{e_c(T)}}{\sum_{j \in N} \frac{x e_{c-1}(T \setminus \{t_j\})}{e_c(T)}} = \frac{e_{c-1}(T \setminus \{t_i\})}{(n - c + 1)e_{c-1}(T)}. \tag{3}$$

Based on the number of model updates on each client, we could formulate staleness involved with client model updates. This is achieved by measuring the expected number of global model updates that occur while a client is training, as follows:

**Corollary 2.** *The expected number of updates given by other clients while client $i$ is participating (e.g., training and communicating for $t_i$ time) is approximated as:*

$$\sum_{j \in N - \{i\}} g(T \setminus \{t_i\}, c - 1, j, t_i) = t_i \frac{(n - c)e_{c-2}(T \setminus \{t_i\})}{e_{c-1}(T \setminus \{t_i\})}. \tag{4}$$

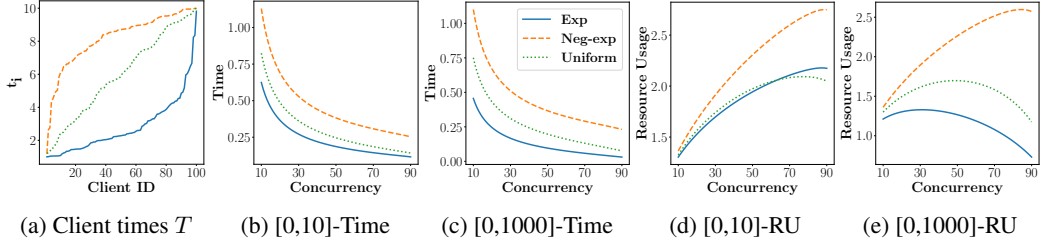

| (a) Client times $T$ | (b) [0,10]-Time | (c) [0,1000]-Time | (d) [0,10]-RU | (e) [0,1000]-RU |

Figure 1: Time and Resource Usage (RU) analysis from the formulation. All subgraphs share the legend of Figure 1c. The results demonstrate the normalized performance of asyncFL, with syncFL's time and RU set as the baseline at 1.0. Figure 1a displays the distribution of three client time sets used for analysis, with each time $t_i$ generated from a scale of [0, 10] and [0, 1000]. Figures 1b and 1d show time and RU for [0, 10] client times, while Figures 1c and 1e show for [0, 1000].

*Let staleness $\tau_i$ denote the number of global model updates that occured during the training and communication time of client $i$. Let $k$ indicate the aggregation goal parameter of asyncFL. The expected value of $\tau_i$:*

$$E[\tau_i] = \frac{Eq.(4)}{k} = \frac{t_i(n-c)e_{c-2}(T \setminus \{t_i\})}{ke_{c-1}(T \setminus \{t_i\})}.$$

*Then, the expected staleness penalty of an asyncFL client:*

$$\sum_{i \in N} \frac{Eq.(3)}{\sqrt{1 + E[\tau_i]}} = \sum_{i \in N} \frac{e_{c-1}(T \setminus \{t_i\})}{(n-c+1)e_{c-1}(T)\sqrt{1 + E[\tau_i]}}. \tag{5}$$

*Note that our formulation uses Eq.(5) to down-weight the global model update of asyncFL.*

Based on the staleness penalty formulation on global model updates, the time is formulated as:

**Proposition 1.** *The time for asyncFL to reach $p$ global model updates is:* $\frac{pke_c(T)}{\sum_{i \in N} \frac{e_{c-1}(T \setminus \{t_i\})}{\sqrt{1+E[\tau_i]}}}.$

*Proof.* The total number of global model updates until time $x$:

$$\frac{\sum_{i \in N} g(T, c, i, x)}{k} = \frac{x}{k} \sum_{i \in N} \frac{e_{c-1}(T \setminus \{t_i\})}{e_c(T)}. \tag{6}$$

Then, the measured global model updates until time $x$ with staleness down-weighting:

$$Eq.(5) \times Eq.(6) = \frac{x}{ke_c(T)} \sum_{i \in N} \frac{e_{c-1}(T \setminus \{t_i\})}{\sqrt{1 + E[\tau_i]}}. \tag{7}$$

Let $p$ global model updates is achieved at time $x$, i.e., Eq.(7) = $p$. Then: $x = \frac{pke_c(T)}{\sum_{i \in N} \frac{e_{c-1}(T \setminus \{t_i\})}{\sqrt{1+E[\tau_i]}}}.$  □

**Resource Usage.** By multiplying $t_i$ by $g(T, c, i, x)$ (i.e., the expected client $i$ updates count until time $x$), we determine the resource usage of client $i$. Then, the total resource usage across all clients:

$$\sum_{i \in N} t_i g(T, c, i, x) = x \sum_{i \in N} t_i \frac{e_{c-1}(T \setminus \{t_i\})}{e_c(T)} = \frac{pk \sum_{i \in N} t_i e_{c-1}(T \setminus \{t_i\})}{\sum_{i \in N} \frac{e_{c-1}(T \setminus \{t_i\})}{\sqrt{1+E[\tau_i]}}}.$$

## 4 CASE STUDY WITH THE FORMULATION

Based on the formulation we derived in Section 3, we conduct analysis on time and resource usage comparing syncFL and asyncFL. We evaluate a toy example involving 100 FL clients, where they have three distinct distributions of client times $T$: (1) *Exp*: Used an exponential distribution to produce client times within the interval [0, 1], with an average of 0.2. This signifies scenarios where only a minority of clients operate at a slower pace. (2) *Neg-exp*: This is achieved by subtracting values from the aforementioned *Exp* distribution from 1 (i.e., 1 - *Exp*), representing a case where

majority of clients are slow. (3) *Uniform*: Clients times are uniformly spread within [0, 1]. Figure 1a demonstrates the example of client times $T$ on the three distributions. To understand the impact of client time scales, we expanded the generated $T$ values into two different ranges: [0, 10] and [0, 1000]. We set aggregation goal parameter as 10 for asyncFL as recommended by Nguyen et al. (2022). We normalize the results by configuring the time and resource usage of syncFL as 1.0. Thus, the asyncFL results demonstrate the relative performance over syncFL. We repeat the generation of $T$ over five different random seeds and analyze on averaged time and resource usage results.

Figures 1b–1e illustrate the time and resource usage results over different concurrency parameters. Although prior studies has asserted a definitive advantage of either asyncFL or syncFL in terms of time and resource usage (Nguyen et al., 2022; Huba et al., 2022), our findings suggest that no single approach always work better than the other. From the time results in Figure 1b and 1c, the majority of our results across three $T$ distributions suggest that asyncFL outpaces syncFL, as evidenced by the results where time $< 1.0$. However, when asyncFL operates on *Neg-exp*, it exhibits a time value exceeding 1.0 at a concurrency of $\approx 10$, signifying that asyncFL is slower than syncFL. In terms of the resource usage, most of the results in Figure 1d and 1e indicate that asyncFL incurs more resource usage than syncFL, showing resource usage $> 1.0$. Yet, asyncFL with *Exp* presents resource usage $< 1.0$ when concurrency $> 75$, being more resource efficient than syncFL.

Our analysis results on the formulation suggest that the time and resource usage of syncFL and asyncFL are shaped by the distribution of the client times $T$ and its scale. Further, these findings hint that the formulation offers valuable insights for FL practitioners, guiding their choices on which approaches and hyperparameters to operate. For instance, if FL clients display a *Neg-exp T* distribution within the range [0, 10], FL practitioners operating on a tight resource budget might naturally gravitate towards syncFL, as the formulation results consistently indicate the resource usage of asyncFL $> 1.0$. More broadly, FL practitioners could pinpoint an optimal concurrency parameter of asyncFL that strikes a right balance between time benefit and additional resource usage over syncFL, by drawing insights from our formulation outcomes.

## 5  PRACTICAL IMPROVEMENTS

In addition to the formulation in Section 3, we make the following two modifications considering realistic scenarios as follows:

**Contribution Scaling on a Client Dataset.** FL clients with heterogeneous data often contribute differently to the global model training (Lai et al., 2021; Shin et al., 2022). We formulate each client's contribution to the training proportional to its dataset size. We define a set $D = \{d_1, d_2, \ldots, d_n\}$, where $d_i$ indicates a dataset size of client $i$. In the formulation, we multiply by $\frac{d_i}{\overline{D}}$ as a contribution factor on each selected client, where $\overline{D} = \frac{\sum_{d_i \in D} d_i}{|D|}$. SyncFL formulation remains identical as expected dataset size of a client sampled at a round is $\overline{D}$. For asyncFL, the formulation is updated by each client item of Eq.(5) by $\frac{d_i}{\overline{D}}$, in addition to the staleness down-weighting.

**Reflecting the Impact of Bias.** In asyncFL, the model gets biased towards faster clients as they contribute more model updates to the global model, while slower clients' updates are down-weighted due to staleness (Tamboli et al., 2023). This issue causes asyncFL to perform more global model updates than syncFL to reach the target accuracy. We define a set $U = \{u_1, u_2, \ldots, u_n\}$, where $u_i$ represents the count of model updates made by client $i$ during asyncFL. The influence of model bias intensifies as the variance of $U$ increases (Zhang et al., 2023), indicating faster clients providing more model updates than slower clients. Thus, we multiply the coefficient of variation of $U$ (denoted as $CV(U)$) by $p$, which we assumed that the target accuracy is achieved after $p$ global model updates in our formulation. Through empirical analysis, we found that multiplying $10 * CV(U) + 1$ at $p$ yields an accurate prediction. Note that adding 1 ensures that p remains unchanged in cases with no variance in U, i.e., CV(U) = 0. We use Eq.(2) to measure $u_i$, which is proportional to $e_{c-1}(T - \{t_i\})$.

## 6  EXPERIMENTS

We conduct experiments to demonstrate how well our formulation predicts the time and resource usage of syncFL and asyncFL under realistic FL scenarios, as follows:

Table 1: Experiment details on five datasets.

|  | FEMNIST | CelebA | Shakespeare | Sent140 | CIFAR-10 |
|---|---|---|---|---|---|
| Num. of clients | 3,000 | 9,343 | 660 | 21,876 | 5,000 |
| Num. of samples | 638,649 | 200,288 | 4,035,372 | 430,707 | 60,000 |
| Target accuracy | 80% | 85% | 50% | 69% | 60% |
| Batch size | 20 | 10 | 4 | 10 | 10 |
| Learning rate | 0.1 | 0.001 | 1 | 0.0003 | 0.1 |
| Model architecture | CNN | CNN | Stacked LSTM | ALBERT | ResNet-18 |

**Datasets.** We run experiments on the following five datasets: FEMNIST (Cohen et al., 2017), CelebA (Liu et al., 2015), Shakespeare (Shakespeare, 2014), Sent140 (Go et al., 2009), and CIFAR-10 (Krizhevsky et al., 2009). Table 1 shows the statistics, model architectures, hyperparameters, and target accuracies on each dataset, which we chose based on previous studies (Caldas et al., 2018; Li et al., 2020; Acar et al., 2021; Charles et al., 2021; Hu et al., 2022; Nguyen et al., 2022; Shin et al., 2022). We trained image classification models on three datasets: FEMNIST contains images of 62 handwritten alphabets and digits from 3,000 clients, and CelebA involves facial images of 9,343 clients. We divided the CIFAR-10 dataset into 5,000 clients as in Nguyen et al. (2022). We also used two natural language-based datasets: we performed next-character prediction on the Shakespeare dataset, which contains speaking lines of 660 roles from Shakespeare's plays. We conduct sentiment analysis on the Sent140 dataset containing Twitter posts of 21,876 clients. We split each client's data into a train, validation, and test set in a ratio of 8:1:1 and measure the validation and test accuracy at every global model update. We allocated IID data to CIFAR-10 dataset clients, but the other four datasets were non-IID as their original form provided by the LEAF framework (Caldas et al., 2018).

**Metrics.** We evaluate metrics in Section 2.2, (1) time, and (2) resource usage on syncFL and asyncFL experiments. The time and resource usage metrics are measured until the test accuracy reaches the target accuracy. We normalize the results by setting up the time and resource usage of syncFL as 1.0 and measure the prediction error by root-mean-square error (RMSE). We further report the accuracy for each experiment, which is a test accuracy at maximal validation accuracy.

**Baselines.** As a baseline method for predicting the time and resource usage of syncFL and asyncFL, we employ the following method that runs FL for a limited number of global model updates and makes predictions based on the mean observed values. We name the baseline methods as *updates-1*, *updates-10*, and *updates-100*, each using observed results from $r$ number of global model updates for prediction where $r$ = 1, 10 and 100.

**Methods.** To emulate realistic training and networking durations of FL clients, we derived client times from a large-scale trace dataset (Yang et al., 2021) that contains on-device training and communication latencies of 136k user smartphones spanning a thousand device models. For each of the five datasets, we generated a client time set $T = \{t_1, \ldots, t_n\}$ whose size $n$ matches the number of clients in the dataset. We use $T$ to predict the metrics on our formulation and measure the prediction error after running syncFL and asyncFL by assigning each $t_i$ on clients. We run up to 5,000 global model updates for FEMNIST, CelebA, and CIFAR-10 datasets; 1,000 for Shakespeare and Sent140 datasets. For asyncFL experiments, concurrency $c$ was set to [10, 25, 50, 100, 200, 500, 1000] (Nguyen et al., 2022). We used [10, 50, 100] aggregation goal parameters $k$ for asyncFL and per-round sampled clients for syncFL. Notably, $k = 10$ consistently yielded minimal time and resource usage across all datasets on both syncFL and asyncFL. We repeated the process five times on different random seeds [0-4], generating different $T$ on each seed, and reported the averaged results.

## 6.1 TIME AND RESOURCE USAGE PREDICTION

Table 2 presents the time and resource usage prediction error on five datasets, averaged from experiments with different concurrency parameters and random seeds. The results indicate our formulation has minimal error among the tested methods in predicting time and resource usage across the five datasets. Our formulation shows 1.08–2.25× and 1.51–5.50× smaller prediction error than baseline methods for the time and resource usage respectively. Our formulation accounts for the inefficiencies of stale model updates in asyncFL, enabling it to outperform baseline methods that treat all asyncFL updates as equivalent to syncFL. Note that our formulation also achieves minimal overhead in performing prediction, obviating the need for test FL rounds unlike the baseline methods.

Table 2: Time and Resource Usage (RU) prediction performance on five datasets, in root-mean-square error (RMSE). Our formulation (*Ours*) is compared with the *Updates-N* baseline methods, which runs FL for $N$ number of global model updates and makes predictions derived from average observed values. Experiments are repeated five times on different random seeds.

| | FEMNIST | | CelebA | | Shakespeare | | Sent140 | | CIFAR-10 | |
|---|---|---|---|---|---|---|---|---|---|---|
| Method | Time | RU | Time | RU | Time | RU | Time | RU | Time | RU |
| Updates-1 | $0.10_{\pm 0.03}$ | $5.00_{\pm 1.92}$ | $0.09_{\pm 0.02}$ | $3.63_{\pm 0.34}$ | $0.17_{\pm 0.04}$ | $6.16_{\pm 0.53}$ | $0.24_{\pm 0.12}$ | $9.18_{\pm 5.65}$ | $0.10_{\pm 0.02}$ | $4.23_{\pm 0.31}$ |
| Updates-10 | $0.11_{\pm 0.02}$ | $4.89_{\pm 1.96}$ | $0.17_{\pm 0.06}$ | $3.46_{\pm 0.35}$ | $0.26_{\pm 0.02}$ | $5.86_{\pm 0.54}$ | $0.20_{\pm 0.11}$ | $9.09_{\pm 5.64}$ | $0.18_{\pm 0.01}$ | $4.15_{\pm 0.30}$ |
| Updates-100 | $0.12_{\pm 0.02}$ | $4.66_{\pm 1.97}$ | $0.18_{\pm 0.06}$ | $3.16_{\pm 0.35}$ | $0.26_{\pm 0.02}$ | $5.60_{\pm 0.55}$ | $0.21_{\pm 0.13}$ | $8.83_{\pm 5.64}$ | $0.20_{\pm 0.01}$ | $3.91_{\pm 0.33}$ |
| **Ours** | $\mathbf{0.08_{\pm 0.02}}$ | $\mathbf{1.95_{\pm 1.21}}$ | $\mathbf{0.08_{\pm 0.04}}$ | $\mathbf{0.66_{\pm 0.26}}$ | $\mathbf{0.13_{\pm 0.01}}$ | $\mathbf{1.58_{\pm 0.24}}$ | $\mathbf{0.14_{\pm 0.10}}$ | $\mathbf{5.85_{\pm 4.11}}$ | $\mathbf{0.09_{\pm 0.02}}$ | $\mathbf{1.77_{\pm 0.33}}$ |

(a) FEMNIST-Time  (b) CelebA-Time  (c) FEMNIST-RU  (d) CelebA-RU

(e) FEMNIST-Updates #  (f) CelebA-Updates #  (g) FEMNIST-Accuracy  (h) CelebA-Accuracy

Figure 2: Experimental results on FEMNIST and CelebA datasets over different concurrency parameters. Figure 2a–2d demonstrates the time and Resource Usage (RU) ground truth values and prediction from *Updates-100* baseline and our formulation. 2e–2f shows the number of global model updates required to reach the target accuracy, and 2g–2h indicates accuracy from each experiment.

Figures 2a–2d illustrate the ground truth and prediction values of the time and resource usage over different concurrency parameters, on FEMNIST and CelebA datasets. The results indicate our formulation better aligns with the ground truth than the *Updates-100* baseline, which especially shows huge discrepancy on resource usage. The baseline, which predicts after running fixed number of global model updates, assumes that both syncFL and asyncFL achieve the target accuracy after identical count of updates; however, as shown in Figures 2e and 2f, asyncFL necessitates a greater number of global model updates to reach the target accuracy than syncFL. The number escalates with bigger concurrency, due to the increasing staleness involved with the global model updates.

Such a staleness factor also impacts the accuracy of asyncFL (Figures 2g and 2h), manifesting decreasing trend at higher concurrency parameters. This is due to the model being biased towards faster clients, given that the updates from slower clients are significantly down-weighted due to their increased staleness (Zhang et al., 2023). In summary, adopting concurrency parameters $\leq 200$ when employing aggregation goal of 10 shows accuracy on par with syncFL across five datasets. We observe that using concurrency $> 200$ offers minimal benefit on time but demands exponentially increasing resource usage as depicted in Figure 2.

**Impact of practical improvements.** We conducted an ablation study to understand the prediction performance brought by each component of *practical improvements* (Section 5): (i) contribution scaling on a client dataset ($D$) and (ii) reflecting the impact of bias ($B$). We refer the original formulation presented in Section 3 as *Orig*, and observe how the prediction error changes when each component is introduced. Figures 3a and 3b report the comparison results of average RMSE for time and resource usage prediction across the datasets. The results show that *Orig*, the formulation without practical improvements, still achieves 1.42-1.54× smaller RMSE on average compared to the baseline method, *Updates-100*. We make two observations: (1) the prediction error drops as components are introduced; (2), $B$ induces more error reduction than $D$. The results imply that training

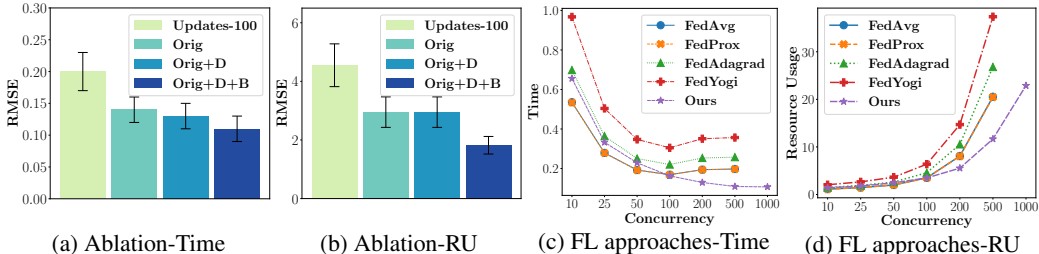

(a) Ablation-Time      (b) Ablation-RU      (c) FL approaches-Time      (d) FL approaches-RU

Figure 3: Experiment results. Figures 3a and 3b depicts the result of an ablation study to compare the predictive error when each element of the *practical improvements* (Section 5) is introduced. This includes scaling contributions based on a client dataset (*D*) and accounting for bias effects (*B*). *Orig* indicates the original formulation from Section 3, and *Updates-100* is the baseline method. Figures 3c and 3d demonstrates how the time and RU result changes on FEMNIST dataset when different FL approaches are applied instead of FedAvg, and how it compares with the formulation.

bias between clients has greater impact on asyncFL than dataset size heterogeneity. Nevertheless, applying *D* to the formulation is still beneficial when FL practitioners deal with FL tasks with large variance on client sample counts.

## 6.2 COMPATIBILITY WITH OTHER FL APPROACHES

We primarily focused on formulating and evaluating the most commonly used FL algorithms for syncFL and asyncFL, which are FedAvg (McMahan et al., 2017) and FedBuff (Nguyen et al., 2022) respectively. However, since the inception of FL, numerous other FL methods and optimizations have been proposed. To understand the impact of these alternative techniques on the prediction error of our formulation, we experimented other widely recognized FL optimization approaches for syncFL in place of FedAvg: a client-side regularizer, *FedProx* (Li et al., 2020), and server-side optimizers, *FedYogi* and *FedAdagrad* (Reddi et al., 2021). We applied recommended parameters for each of the approaches: $\mu = 0.001$ for FedProx, $\{\beta_1, \beta_2, \eta, \tau\} = \{0.9, 0.99, 0.01, 0.001\}$ for FedYogi and FedAdagrad.

Figures 3c and 3d demonstrate the time and resource usage results compared to our formulation over different concurrency parameters on FEMNIST dataset. While substituting different methods for FedAvg leads to different prediction errors, the overall trend observed across different concurrency parameters and our formulation remains consistent. This indicates the potential of our formulation to predict trends even when alternative FL approaches are implemented. Moreover, we hypothesize that our formulation's prediction error on other FL approaches would decrease when analogous optimization is applied on asyncFL. As such optimization techniques are specifically designed for syncFL, directly applying such approaches on asyncFL is non-trivial. Additionally, refining our formulation to incorporate various optimization techniques for both syncFL and asyncFL is what we plan to explore as future work.

## 7 CONCLUSION

We present a formulation that quantifies the time and resource usage of synchronous and asynchronous FL (syncFL and asyncFL), given the training and communicating speed of participating clients. Our formulation factors in the inefficiencies of asyncFL due to stale model updates, facilitating a more precise comparison with syncFL in achieving the same objectives. Our analysis on the formulation highlights that the distribution and scale of clients' training and communication time influences the time and resource usage of both approaches. Contrary to prior findings, our formulation reveals that neither approach consistently outperforms the other in terms of time and resource usage. Our evaluation on five datasets demonstrates that the formulation accurately predicts the relative time and resource usage of syncFL and asyncFL, with 5.5× smaller root-mean-square error (RMSE) than the baseline methods. We believe our formulation will provide valuable insights on the efficiency of syncFL and asyncFL for FL practitioners, guiding them to make informed decisions between the two approaches based on their resource constraints.

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

## A    PROOF OF LEMMA 1

*Proof.* We prove by induction that $f(T, c, i) = \frac{t_i e_{c-1}(T \setminus \{t_i\})}{e_c(T)}$, where function $f(T, c, i)$ denotes the portion of time in which client $i$ participated in asyncFL ($0 \le f(T, c, i) \le 1$). $e_c(T)$ is an elementary symmetric polynomial defined as: $e_c(T) = \Sigma_{i_1 < i_2 < \dots < i_c} t_{i_1} t_{i_2} \dots t_{i_c}$, where $t_{i_1}, t_{i_2}, \dots, t_{i_c} \in T$. Refer to Section 3 for other notations.

**Basis.**    If $T = \{t_1\}$, then $f(T, 1, 1) = 1$.

If $T = \{t_1, t_2\}$, then $f(T, 1, 1) = \frac{t_1}{t_1 + t_2}$, $f(T, 1, 2) = \frac{t_2}{t_1 + t_2}$, $f(T, 2, 1) = 1$, $f(T, 2, 2) = 1$.

For $T = \{t_1, \dots, t_m\}$ where $m \ge 1$, $f(T, 1, 1) = \frac{t_i}{\Sigma T}$ and $f(T, m, i) = 1$ for $i \in 1, \dots, m$.

**Hypothesis.**    Assume $f(T, c, i) = \frac{t_i e_{c-1}(T \setminus \{t_i\})}{e_c(T)}$ holds for any set $T$ satisfying $|T| \le n$, where $c \le |T|$, $t_i \in T$.

**Inductive Step.**    Let $T = \{t_1, t_2, \dots, t_n\}$, and $T' = T + t_{n+1}$.

For $t_i, t_j \in T'$ and $c < |T'|$, the following holds:

$$f(T', c, i) = f(T', c, j)f(T' \setminus \{t_j\}, c - 1, i) + (1 - f(T', c, j))f(T' \setminus \{t_j\}, c, i). \qquad (8)$$
$$f(T', c, j) = f(T', c, i)f(T' \setminus \{t_i\}, c - 1, j) + (1 - f(T', c, i))f(T' \setminus \{t_i\}, c, j). \qquad (9)$$

Calculating $f(T', c, i)$ in Eq.(8) indicates the summation of the following two cases: (1) $t_i$ is participating with $t_j$: $f(T', c, j)f(T' \setminus \{t_j\}, c-1, i)$, and (2) $t_i$ is participating while $t_j$ is not participating: $(1 - f(T', c, j))f(T' \setminus \{t_j\}, c, i)$. Substituting $t_i$ as $t_j$ and $t_j$ as $t_i$ results in Eq.(9).

Using the recursion property of elementary symmetric polynomials (for $1 \le c \le |T|$):

$$e_c(T) = t_i e_{c-1}(T \setminus \{t_i\}) + e_c(T \setminus \{t_i\}),$$

and using our inductive hypothesis, let:

$$
\begin{aligned}
y_1 = f(T' \setminus \{t_j\}, c - 1, i) &= \frac{t_i e_{c-2}(T' \setminus \{t_i, t_j\})}{e_{c-1}(T' \setminus \{t_j\})} \\
&= \frac{t_i e_{c-2}(T' \setminus \{t_i, t_j\})}{t_i e_{c-2}(T' \setminus \{t_i, t_j\}) + e_{c-1}(T' \setminus \{t_i, t_j\})}, \\
y_2 = f(T' \setminus \{t_j\}, c, i) &= \frac{t_i e_{c-1}(T' \setminus \{t_i, t_j\})}{e_c(T' \setminus \{t_j\})} \\
&= \frac{t_i e_{c-1}(T' \setminus \{t_i, t_j\})}{t_i e_{c-1}(T' \setminus \{t_i, t_j\}) + e_c(T' \setminus \{t_i, t_j\})}, \\
y_3 = f(T' \setminus \{t_i\}, c - 1, j) &= \frac{t_j e_{c-2}(T' \setminus \{t_i, t_j\})}{e_{c-1}(T' \setminus \{t_i\})} \\
&= \frac{t_j e_{c-2}(T' \setminus \{t_i, t_j\})}{t_j e_{c-2}(T' \setminus \{t_i, t_j\}) + e_{c-1}(T' \setminus \{t_i, t_j\})}, \\
y_4 = f(T' \setminus \{t_i\}, c, j) &= \frac{t_j e_{c-1}(T' \setminus \{t_i, t_j\})}{e_c(T' \setminus \{t_i\})} \\
&= \frac{t_j e_{c-1}(T' \setminus \{t_i, t_j\})}{t_j e_{c-1}(T' \setminus \{t_i, t_j\}) + e_c(T' \setminus \{t_i, t_j\})}, \\
\alpha &= f(T', c, i), \\
\beta &= f(T', c, j).
\end{aligned}
$$

Using $y_1$, $y_2$, $y_3$, $y_4$, $\alpha$, and $\beta$, Eq.(8) and (9) each becomes the following:

$$
\begin{aligned}
\alpha &= \beta y_1 + (1 - \beta)y_2, \\
\beta &= \alpha y_3 + (1 - \alpha)y_4.
\end{aligned}
$$

Substituting $\beta$ with an expression of $\alpha$ results in:

$$\alpha = (\alpha y_3 + (1 - \alpha)y_4)y_1 + (1 - (\alpha y_3 + (1 - \alpha)y_4))y_2,$$

$$\alpha = \frac{y_1 y_4 + y_2 - y_2 y_4}{1 - y_1 y_3 + y_1 y_4 + y_2 y_3 - y_2 y_4}. \tag{10}$$

Let $z_1 = e_c(T' \setminus \{t_i, t_j\})$, $z_2 = e_{c-1}(T' \setminus \{t_i, t_j\})$, $z_3 = e_{c-2}(T' \setminus \{t_i, t_j\})$.

Then, numerator of Eq.(10) becomes:

$$
y_1 y_4 + y_2 - y_2 y_4 = \frac{t_i z_3}{t_i z_3 + z_2}\frac{t_j z_2}{t_j z_2 + z_1} + \frac{t_i z_2}{t_i z_2 + z_1} - \frac{t_i z_2}{t_i z_2 + z_1}\frac{t_j z_2}{t_j z_2 + z_1}
$$
$$
= \frac{(t_i t_j z_2 z_3)(t_i z_2 + z_1) + t_i z_2(t_i z_3 + z_2)(t_j z_2 + z_1) - t_i t_j z_2^2(t_i z_3 + z_2)}{(t_i z_3 + z_2)(t_i z_2 + z_1)(t_j z_2 + z_1)}
$$
$$
= \frac{t_i z_2(t_i t_j z_2 z_3 + t_i z_1 z_3 + t_j z_1 z_3 + z_1 z_2)}{(t_i z_3 + z_2)(t_i z_2 + z_1)(t_j z_2 + z_1)}.
$$

The denominator of Eq.(10) is:

$$
1 - y_1 y_3 + y_1 y_4 + y_2 y_3 - y_2 y_4
$$
$$
= 1 - \frac{t_i z_3}{t_i z_3 + z_2}\frac{t_j z_3}{t_j z_3 + z_2} + \frac{t_i z_3}{t_i z_3 + z_2}\frac{t_j z_2}{t_j z_2 + z_1} + \frac{t_i z_2}{t_i z_2 + z_1}\frac{t_j z_3}{t_j z_3 + z_2} - \frac{t_i z_2}{t_i z_2 + z_1}\frac{t_j z_2}{t_j z_2 + z_1}
$$
$$
= \frac{z_2(t_i t_j z_3 + t_i z_2 + t_j z_2 + z_1)(t_i t_j z_2 z_3 + t_i z_1 z_3 + t_j z_1 z_3 + z_1 z_2)}{(t_i z_3 + z_2)(t_j z_3 + z_2)(t_i z_2 + z_1)(t_j z_2 + z_1)}.
$$

Thus, Eq.(10) becomes:

$$
f(T', c, i) = \alpha
$$
$$
= \frac{y_1 y_4 + y_2 - y_2 y_4}{1 - y_1 y_3 + y_1 y_4 + y_2 y_3 - y_2 y_4}
$$
$$
= \frac{\frac{t_i z_2(t_i t_j z_2 z_3 + t_i z_1 z_3 + t_j z_1 z_3 + z_1 z_2)}{(t_i z_3 + z_2)(t_i z_2 + z_1)(t_j z_2 + z_1)}}{\frac{z_2(t_i t_j z_3 + t_i z_2 + t_j z_2 + z_1)(t_i t_j z_2 z_3 + t_i z_1 z_3 + t_j z_1 z_3 + z_1 z_2)}{(t_i z_3 + z_2)(t_j z_3 + z_2)(t_i z_2 + z_1)(t_j z_2 + z_1)}}
$$
$$
= \frac{t_i(t_j z_3 + z_2)}{t_i t_j z_3 + t_i z_2 + t_j z_2 + z_1}
$$
$$
= \frac{t_i(t_j e_{c-2}(T' \setminus \{t_i, t_j\}) + e_{c-1}(T' \setminus \{t_i, t_j\}))}{t_i(t_j e_{c-2}(T' \setminus \{t_i, t_j\}) + e_{c-1}(T' \setminus \{t_i, t_j\})) + t_j e_{c-1}(T' \setminus \{t_i, t_j\}) + e_c(T' \setminus \{t_i, t_j\})}
$$
$$
= \frac{t_i(e_{c-1}(T' \setminus \{t_i\}))}{t_i e_{c-1}(T' \setminus \{t_i\}) + e_c(T' \setminus \{t_i\})}
$$
$$
= \frac{t_i(e_{c-1}(T' \setminus \{t_i\}))}{e_c(T')}.
$$

The inductive hypothesis holds true for $f(T', c, i)$, establishing the inductive step for $c \leq |T|$ and any $i$ which $t_i \in T'$. $|T'|$ is the only possible $c > |T|$ for $f(T', c, i)$, as we defined $f$ to have $c \leq$ cardinality of the set given as a first parameter. As noted in the basis step of induction, $f(T', |T'|, i) = 1$. This proves the hypothesis holds for arbitrary set $T$, $c \leq |T|$, and any $i$ which $t_i \in T$. $\qquad \square$

