# OpenReview forum: "Are We in (A)Sync?: Guidance for Efficient Federated Learning"
_ICLR.cc/2024/Conference — Submitted to ICLR 2024_

### Official Review · Reviewer_eseE · 2023-10-28

**Soundness:** 2 fair
**Presentation:** 1 poor
**Contribution:** 2 fair
**Rating:** 6
**Confidence:** 3

**Summary:**

This paper proposes a model to formulate the time consumption and resource usage in asynchronous FL methods and tries to use this model to understand the advantage of the asynchronous FL method compared to the synchronous FL method. The idea is interesting and promising, and the topic is meaningful.

**Strengths:**

The idea is interesting and promising, and the topic is meaningful. The experiment results show that the proposed formulation well approximates the actual time consumption and resource allocation. Overall, I like the idea of this paper, but wish the authors could clarify my concerns regarding the technical results. I'm willing to improve my score if the authors can address some of my concerns.

**Weaknesses:**

The technical part of this paper is not well-written and difficult to understand to the reviewer.

1. It's difficult for me to understand the real meaning of $f(T, c, i)$. The authors claimed that $f(T, c, i)$ is the portion of time client $i$ participated in AsyncFL. If the training process takes time $A$ and during the $A$ time interval, client $i$ participated in the training process for $B_i$ time units, then I would think that $f(T,c, i)=B_i/A$. However, with this definition, I cannot understand why in Appendix A, $f(T,2,1)=f(T,2,2)=1$ when $T=$\{$ t_1,t_2$ \}. Moreover, I also don't understand why $f(T,c, i)$ is irrelevant with $k$.

2. The expression of some terminologies is not accurate. For example, the number of updates given by other clients during the training and communication of node $i$ is not precisely the quantity in Eq (4), and the quantity in Eq (4) only tells the of updates given by other clients during the $0$th update and the $1$th update of node $i$.

3. I understand that for simplicity, the authors treat delays or the number of updates as continuous variables, rather than discrete variables. However, since these quantities themselves are discrete, the authors should at least mention this in the paper.

**Questions:**

The authors also provide a formula for delay prediction (above Eq (5)), which is of interest to many researchers. Therefore, can the authors compare the actual delay distribution with the predicted one by experiments? It would be great if you could provide such a comparison even after the rebuttal and in the final version.

---

> ### Author Response · Authors · 2023-11-19
> **Response to Reviewer eseE**
>
> We truly appreciate your thoughtful comments and valuable suggestions. Our responses are as follows:
>
> ---
>
> **Comment 1: Meaning of $f(T,c,i)$ and its relevance with respect to $k$**
>
> Thank you very much for your detailed question on our formulation. Your understanding is correct: if the client $i$ participated in asyncFL for $B_i$ time while the whole training process takes time $A$, then $f(T,c,i) = \frac{B_i}{A}$.
>
> We would like to confirm that when $T = \{t_1, t_2\}$, $f(T,2,1) = f(T,2,2) = 1$ is correct, $f(T,2,1)$ and $f(T,2,2)$ indicates the portion of time which client 1 and client 2 each participated in asyncFL when concurrency value $c = 2$. As concurrency $= 2$ when the total number of clients $= 2$, client 1 and client 2 should continuously participate in asyncFL during the whole process to satisfy the concurrency parameter, which makes $f(T,2,1) = f(T,2,2) = 1$. We assume the time required for the server to aggregate updates and to sample new clients are zero for simplicity.
>
> Moreover, $f(T,c,i)$ might depend on $k$ if a client is restricted to contributing more than once to the global model updates, a situation that occurs when $k$ client updates are buffered. However, our formulation is constructed without such restriction, making $k$ irrelevant to $f(T,c,i)$.
>
> We will ensure to present the above in our revised manuscript, providing more details on the assumptions we made and clarifying the explanation of our formulation.
>
> ---
>
> **Comment 2: Inaccurate expression of terminologies**
>
> Thank you for pointing out this oversight. You are correct in noting that Eq. (4) does not precisely calculate the expected number of updates by other clients while client i is training and communicating. However, it offers a close approximation, as evidenced by our formulation based on Eq. (4), which closely predicts the actual time and resource usage observed in FL experiments. This is further supported by our delay prediction results directly derived from Eq. (4), as mentioned below in comment 4. We recognize the need to clarify this in our manuscript and either plan to revise Eq. (4) or explicitly state its approximate nature in the updated version.
>
> ---
>
> **Comment 3: Treating discrete variables as continuous variables**
>
> Thank you for bringing this to our attention. The number of updates and the delays (staleness) we mentioned in Section 3 are all expected values. We will ensure to update our inaccurate notations regarding the continuous or discrete nature of variables in our updated manuscript.
>
> ---
>
> **Comment 4 (Question): Comparison on formulated delay prediction and actual value**
>
> Thank you for your suggestion. We conducted a delay prediction based on the equation above Eq. (5), by comparing it with the actual delay values from the experiment on the FEMNIST dataset as follows:
>
> | concurrencies       | 10    | 25    | 50    | 100    | 200    | 500    | 1000    |
> |---------------------|-------|-------|-------|--------|--------|--------|---------|
> | FEMNIST-Formulation | 0.901 | 2.406 | 4.929 | 10.022 | 20.402 | 53.115 | 113.129 |
> | FEMNIST-Experiment  | 0.900 | 2.399 | 4.896 | 9.885  | 19.845 | 49.546 | 98.503  |
>
> The results indicate that our formulation aligns closely with the actual delay values observed in the FL experiment. However, we observed an increasing discrepancy at higher concurrency levels, likely stemming from the errors in Eq. (4) addressed in comment 2. We expect such an error to decrease as we suggest an accurate formulation for Eq. (4). In our revised manuscript, we plan to include results on delay prediction across all datasets used in our study.

---

> > ### Comment · Reviewer_eseE · 2023-11-20
> > **I agree to raise my score because the authors addressed some of my issues.**
> >
> > I agree to raise my score because the authors addressed some of my issues. Specifically, they demonstrate that the delay formula (4) they derived is relatively tight.

---

> > > ### Author Response · Authors · 2023-11-22
> > > **To reviewer eseE**
> > >
> > > We truly appreciate that you raised the score, and we are very happy that our responses have addressed the issues. Thank you very much again for your constructive feedback on our paper.

---

### Official Review · Reviewer_F9nE · 2023-10-29

**Soundness:** 2 fair
**Presentation:** 2 fair
**Contribution:** 2 fair
**Rating:** 3
**Confidence:** 4

**Summary:**

AsyncFL allows the server to exchange models with available clients continuously, enhancing the resource utilization. Given the training and communication speed of participating clients, this paper presents a formulation of time and resource usage on syncFL and asyncFL. The proposed formulation weights asyncFL against its inefficiencies stemming from stale model updates, enabling more accurate comparison to syncFL. This paper reveals that no single approach always works better than the other regarding time and resource usage.

**Strengths:**

1. The finding that "neither syncFL nor asyncFL universally outperforms the other in terms of time and resource usage" is interesting.
2. The studied problem is timely and may have practical influences.

**Weaknesses:**

1. Lemma 1, Corollary 1, 2 and Proposition 1 consider the participating time and resource usage. However, they do not consider the model training, loss functions, data heterogeneity, etc. Thus, it is hard to say the proposition can be utilized into FL.
2. Non-IID data distribution widely exists in FL. However, experiments only consider IID data distribution.
3. The presentation of experiment results is not clear. What does the proposed formulation mean when compared with other FL algorithms?

**Questions:**

See weaknesses.

---

> ### Author Response · Authors · 2023-11-19
> **Response to Reviewer F9nE**
>
> We greatly appreciate your thoughtful comments and feedback. Our replies are as follows:
>
> ---
>
> **Comment 1: Formulation not considering training, loss functions, or data heterogeneity**
>
> Thank you for highlighting an important aspect to consider regarding our formulation. Incorporating model convergence, training specific, and data characteristics would yield more theoretically precise results. However, we instead approached formulating with an assumption that syncFL and asyncFL achieve the target accuracy after p global model updates. Our empirical results in Section 6 demonstrate that this simplified formulation also effectively predicts the actual values observed in FL runs across five datasets spanning vision and NLP domains that involve up to 21,876 clients. We plan to improve our theoretical formulation based on the elements you mentioned as a future work.
>
> ---
>
> **Comment 2: Experiments considering IID data distribution only**
>
> Thank you for emphasizing the importance of non-IID data distribution in federated learning. In our experiments, while we allocated IID data to clients within the CIFAR-10 dataset, the other four datasets (FEMNIST, CelebA, Shakespeare, Sent140) employed were distinctly non-IID. We utilized these four datasets in their original form, as provided by the LEAF federated learning framework [1], where they inherently exhibit non-IID characteristics with heterogeneous data quantity and class distributions.
>
> In our revised manuscript, we will include details on non-IID characteristics of experimented datasets.
>
> ---
>
> **Comment 3: Meaning of proposed formulation compared to other algorithms**
>
> We interpreted your comment (item 3 under weaknesses) as about the experimental results shown in Figures 3c and 3d and have responded accordingly below. If your comment was about a different aspect, please inform us.
>
> We built our formulation on the prevalent syncFL and asyncFL algorithms (FedAvg and FedBuff). In experiments on Figures 3c and 3d, we aimed to assess whether our formulation provides insight into the time and resource usage when other widely used FL optimization algorithms are applied, such as FedProx, FedYogi, and FedAdagrad. Our results indicate that employing our formulation with these algorithms may increase the prediction error. Still, the general trend in time and resource usage over different concurrency parameters remains consistent between the formulation and other FL algorithms.
>
> We empirically showed our formulation’s applicability in predicting time and resource usage trends over different concurrency parameters. This could potentially assist FL practitioners in choosing the most suitable approach (asyncFL / syncFL) or parameter for different FL algorithms.
>
> ---
>
> [1] Caldas, Sebastian, et al. "Leaf: A benchmark for federated settings." arXiv preprint arXiv:1812.01097 (2018).

---

> ### Comment · Reviewer_F9nE · 2023-11-20
> **Thanks for your respones**
>
> Thanks for your respones, I decide to keep my scores based on following comments:
> 1. The analysis considering training, loss functions, or data heterogeneity is important.
> 2. I checked the revision, the details of the dataset partition are not provided.

---

> ### Author Response · Authors · 2023-11-22
> **To reviewer F9nE**
>
> Thank you very much again for your valuable feedback.
>
> We fully acknowledge the importance of considering training, loss functions, and data heterogeneity in our research. While we regret not including these in the current discussion due to time constraints, we would like to stress that our simplified formulation accurately predicts the actual values observed in FL executions.
>
> In the paper, we added an explanation that four out of five used datasets were non-IID as originally provided by LEAF. We opted not to include extensive details of the dataset in our paper, as the datasets we used are well-established in the field and the details are covered in prior literature.

---

### Official Review · Reviewer_pJDG · 2023-11-02

**Soundness:** 3 good
**Presentation:** 3 good
**Contribution:** 3 good
**Rating:** 3
**Confidence:** 4

**Summary:**

This paper aims to show that neither synchronous (syncFL) nor asynchronous (asyncFL) Federated Learning (FL) approaches can be deemed to be definitively superior over the other in regards to reducing time and resource consumption, thus invalidating the previous findings which showed one's superiority over the other.

**Strengths:**

++ The paper makes a novel observation: the current works that compare syncFL and asyncFL concerning their time and resource consumption contradict each other––some works claim that asyncFL is better than syncFL, and others claim the opposite. The authors settle this argument by making a novel statement that there is no definitive winner among the two.

++ The authors have thoroughly examined the related works, and identified flaws and contradictions among those works.

++ It introduces novel formulations to determine the time and resource usage till the target accuracy is achieved. The formulation accounts for stale model updates in asyncFL which allows it to outperform baseline models.

**Weaknesses:**

-- Lack of comparison with state-of-the-art approaches for time & resource measurement and estimation. The paper only compares with the Updates-N baseline methods.

-- The abstraction of resource usage in FL is over-simplified since real-world FL systems rely on multiple types of resources with heterogeneous characteristics.

-- The authors' formulations for time and resource consumption make strong assumptions.  The formulations for time and resource consumption assume that the syncFL and asyncFL reach the target accuracy after p rounds. How do we determine the number (p) of updates until target accuracy is achieved? In addition, the authors' formulation assumes that the time (T = {t_1,t_2,...,t_n}) required by the clients to download, train and upload the model weights are constant across training rounds. The assumption of T being constant across rounds may not reflect reality because a client model can be faster in certain rounds and slower in others.

-- In section 4, the authors conclude that, based on their formulations, neither syncFL nor asyncFL can be deemed to be definitively superior to the other. The authors have used their formulation to demonstrate that neither syncFL nor asyncFL can be deemed to be definitively superior to the other. They should verify this using actual time and resource usage values.

**Questions:**

1. In section 5, under "Reflecting the Impact of Bias", the authors claim that $10*CV(U)+1$ at $p$ yields an accurate prediction. The authors should justify this in the paper.

2. How do the experiments support the authors' argument–––neither syncFL nor asyncFL can be deemed to be definitively superior to the other? The authors have shown that their approach of determining resource and time utilization for asyncFL closely approximates the actual values, however since this does not establish a connection with previous works that have opposing views, it does not invalidate the previous authors' works which determines that either asyncFL is better than syncFL, or vice versa. Figures 3c and 3d do not justify that the authors' formulations are also accurate when predicting time and resource usage for other aggregation schemes. Those figures do not compare the formulations' predictions of time and resource usage to real ones, instead, they simply show the predictions when using the authors' formulations.

Writing Issues:

* Figure 1 has missing legends, making it incomprehensible to the readers. The figure is critical to the paper as it intends to show that neither syncFL nor asyncFL approaches can be deemed to be definitively superior to the other in regards to reducing time and resource consumption. The authors mention that Figure 1 is a comparison of asyncFL and syncFL in terms of their resource and time utilization, however, it appears that the figure is incomplete and does not compare the two approaches.

* What is D-bar in section 5, under "Contribution Scaling on a Client Dataset"?

---

> ### Author Response · Authors · 2023-11-19
> **Response to Reviewer pJDG (Part 1)**
>
> We are deeply grateful for your thoughtful comments and feedback. In our revised manuscript, we intend to address these points as follows:
>
> ---
>
> **Comment 1: Lack of comparison with state-of-the-art approaches**
>
> We appreciate your question regarding the comparison with state-of-the-art approaches. To our knowledge, no existing literature proposes to predict the relative time or resource usage between syncFL and asyncFL. Thus, we could only compare with a basic baseline, “Updates-$N$,” which predicts after observing the metrics for $N$ rounds. In our revised manuscript, we plan to introduce an additional baseline that predicts based on the observed metrics from initial FL runs reaching a low target accuracy (e.g., 10%). Please note that these baseline methods require sample FL runs for each configuration to make predictions, whereas our formulation uniquely predicts the metrics without requiring any FL runs.
>
> ---
>
> **Comment 2: Resource usage in FL is over-simplified**
>
> Thank you for questioning our abstraction of resource usage as the cumulative time for on-device training and model weight communication. We acknowledge the variety of resources in real-world FL systems and the value of examining each type separately. Although capable of presenting our resource usage formulation for each type, we opted for a time-based abstraction to simplify our prediction experiments on actual values. The time units metric is proportional to various resource types such as energy consumption [1]. It allows us to avoid fine-grained measurements (e.g., power measurements), which are challenging to accurately simulate at scale. Our choice of a time-based abstraction is based on Abdelmoniem et al. [2], who explored the resource efficiency of federated learning systems.
>
> We will ensure to explain our choice of resource usage abstraction in our revised manuscript.
>
> ---
>
> **Comment 3: Assumption in time and resource usage formulations**
>
> We acknowledge that our formulation was built on an assumption that syncFL and asyncFL achieve the target accuracy after p global model updates. However, we would like to note that in actual FL runs, the number of required global model updates observed in both approaches is similar while having different time and resource usage, as shown below:
>
> | Datasets | SyncFL | AsyncFL c:10 | AsyncFL c:25 | AsyncFL c:50 | AsyncFL c:100 |
> |----------|--------|--------------|--------------|--------------|---------------|
> | FEMNIST  | 131    | 115          | 116          | 124          | 164           |
> | Celeba   | 518    | 649          | 574          | 569          | 567           |
>
> The numbers of asyncFL global model updates in the table are adjusted with the staleness penalty as proposed in Section 3. We observed that the required global model updates for asyncFL are generally comparable to those needed for syncFL. This suggests that our formulation could be leveraged to predict time and resource usage of actual FL runs, as we evidenced in our experiments in Section 6.
>
> Addressing your question on client times varying over time, we conducted an additional experiment to compare the prediction errors in scenarios where client times are either changing or remaining constant. To simulate dynamically changing client times, we employed a methodology from a previous study [3] that simulated client training and communication times based on real-world user traces from 136,000 smartphones. In this method, each client is assigned a mean and standard deviation for its time, with the client's time being sampled from a normal distribution each time it is selected. For experiments where client times are constant, the mean value was always used. Our experiments on the FEMNIST dataset, using the same hyperparameters as in Section 6, generated the following results:
>
> | Methods              | Time RMSE  | Resource usage RMSE |
> |----------------------|------------|---------------------|
> | Static client times  | 0.07+-0.01 | 0.50+-0.09          |
> | Dynamic client times | 0.06+-0.01 | 0.50+-0.17          |
>
> These results show that the prediction accuracy is remarkably similar in both scenarios. We surmise that the minimal difference arises as the variations in client times in the dynamic case tend to average out over numerous global model updates in FL. This finding implies that our formulation would be effective in real-world conditions. We intend to incorporate these findings on dynamic client times in our revised manuscript, extending the experiment across all datasets used in our study.
>
> ---
>
> [1] Li, Li, et al. "SmartPC: Hierarchical pace control in real-time federated learning system." IEEE Real-Time Systems Symposium, 2019.
>
> [2] Abdelmoniem, Ahmed M., et al. "REFL: Resource-Efficient Federated Learning." The Eighteenth European Conference on Computer Systems, 2023.
>
> [3] Yang, Chengxu, et al. "Characterizing impacts of heterogeneity in federated learning upon large-scale smartphone data." The Web Conference, 2021.

---

> > ### Author Response · Authors · 2023-11-19
> > **Response to Reviewer pJDG (Part 2)**
> >
> > **Comment 4: Verification of findings in Section 4**
> >
> > Thank you for pointing out this oversight. To substantiate our claim in Section 4 that syncFL and asyncFL do not definitively outperform each other, we conducted an additional experiment on the CIFAR-10 dataset with 100 clients, measuring the time and resource usage required to reach the target accuracy (50%). We simulated the training and communication times of each client using the same distributions (Exp, NegExp, Uniform) and scales ([0, 10] and [0, 1000]) as in Figure 1, with other hyperparameters matching those described in Section 6. The table below displays results for NegExp-[0,10] and Exp-[0,1000], which we used to support our argument in Section 4:
> >
> > | Distribution-Scale | Metrics        | AsyncFL c:10 | AsyncFL c:25 | AsyncFL c:50 | AsyncFL c:75 | AsyncFL c:100 |
> > |--------------------|----------------|--------------|--------------|--------------|--------------|---------------|
> > | NegExp-[0,10]      | Time           | 1.130        | 0.847        | 5.273        | 7.620        | X             |
> > | NegExp-[0,10]      | Resource Usage | 1.372        | 2.551        | 32.262       | 70.019       | X             |
> > | Exp-[0,1000]       | Time           | 0.626        | 0.267        | 0.156        | 0.101        | 0.030         |
> > | Exp-[0,1000]       | Resource Usage | 1.463        | 1.530        | 1.743        | 1.650        | 0.532         |
> >
> > The table shows the results of asyncFL runs on different hyperparameters, where the metrics are normalized by configuring the time and resource usage of syncFL as 1.0. The results verify our argument:
> > 1) Time: AsyncFL is generally faster than syncFL by showing < 1.0 time, but for NegExp-[0,10], asyncFL shows slower results than syncFL on several concurrencies.
> > 2) Resource usage: AsyncFL is mostly less efficient than syncFL with > 1.0 resource usage across experiments, except for Exp-[0,1000] experiment with concurrency of 100, which takes <1.0 resource usage.
> >
> > We plan to add such an experimental verification in our revised manuscript.
> >
> > ---
> >
> > **Comment 5 (Question): Justification of claim in Section 5**
> >
> > In Section 5, we proposed multiplying $10 * CV(U) + 1$ for the value of $p$ to address the bias in asyncFL towards faster clients, with U being the set of model update counts from individual clients. This formula penalizes scenarios where faster clients contribute disproportionately more updates, increasing the variance in $U$. We chose to multiply the coefficient of variation of $U$ ($CV(U)$) by 10 based on empirical evidence that it enhances prediction accuracy. Adding 1 ensures that $p$ remains unchanged in cases with no variance in $U$, i.e., $CV(U) = 0$. In our updated paper, we will include the above justifications and the empirical evidence on choosing the multiplication factor of $CV(U)$.
> >
> > ---
> >
> > **Comment 6 (Question): Connectivity between the experiment and the claim**
> >
> > In our submitted paper, we sought to validate our argument in Section 4 that neither syncFL nor asyncFL definitively outperforms the other. The absence of legends in Figure 1 and the lack of experimental verification in Section 4, focusing solely on formulation-based results, might have caused confusion. The experiments in Section 6 were designed not to validate this argument but to demonstrate that our formulation closely approximates actual values. Regarding Figures 3c and 3d, our goal was to determine if our formulation could provide any insight into the actual time and resource usage of different FL optimization algorithms. Thus, the results in Figures 3c and 3d were all experimental results except the plot annotated as “Ours.” We plan to provide additional clarifications in our revised manuscript, which we hope will address your question effectively.
> >
> > ---
> >
> > **Comment 7: Writing issues**
> >
> > Thank you for pointing out the issues in our submitted paper. Figure 1 compares syncFL and asyncFL by normalizing the asyncFL results across various concurrency parameters, with syncFL's time and resource usage set as the baseline at 1.0. Thus, the results demonstrate the relative performance of asyncFL compared to syncFL, based on whether the value is bigger or smaller than 1.0. We will clarify this in the figure caption and add the legends.
> >
> > The D-bar in Section 5 under “Contribution Scaling on a Client Dataset” indicates the mean value of a set $D$. We acknowledge the need for clarifying our notations and will revise our manuscript accordingly.

---

### Meta-Review · Area_Chair_qxjc · 2023-12-14

**Metareview:**

I have read the reviews and also have read the paper. Unfortunately, the approach used in the paper to compare asynchronous and synchronous FL methods is not sound. One of the key issues is a simplifying assumption that the two types of methods can be meaningfully compared on the basis of reaching same number of global updates. This is an unsubstantiated assumption, and the findings of the paper rely on them. I can't believe the results obtained in the paper if I do not believe in the assumption they all stem from.

The inner workings of synchronous and asynchronous methods is much more complicated, as can be seen by inspecting the theoretical results obtained in the two cited papers: Koloskova et al and Mishchenko et al. These papers show that in certain regimes, asynchronous SGD is always superior to minibatch SGD. I do not understand why the authors dud not build upon the break-through findings of these amazing papers.

Furthermore, the work failed to cite the recent work "Optimal time complexities of parallel stochastic optimization methods under a fixed computation model" (NeuriPS 2023) which build upon the aforementioned works and develops the first provably optimal parallel SGD method. The method turns out to be half synchronous and half asynchronous. So, this paper in essence asks a very similar question: are synchronous methods or asynchronous methods better? And the key finding is the discovery of a new hybrid method that is provably better than previous synchronous and asynchronous SGD methods; and moreover, is un-improvable. This work is not cited, even though it seems to be perhaps the most relevant work to the submission. Unlike the submission, the NeurIPS 2023 work offers rigorous bounds under standard assumptions.

When reading the paper, I had the feeling that the key contribution is not only questionable due to unjustified assumptions, but also very simple to execute. To me, the results looked rather trivial, and vert simple to obtain. I would expect much more from an ICLR paper.

Finally, the reviewers are mostly inclined towards rejection. I have read the reasons for rejection and acceptance, but in my mind, the reasons for rejection far outweigh the reasons for acceptance. Combined with my own reading of the paper, I have no choice but to recommend rejection.

**Justification For Why Not Higher Score:**

The paper can't be accepted. It is based on unjustified strong assumptions, misses the most prior relevant paper to theirs, and generally does not read like a strong contribution. The approach looks overly simplistic and analysis almost trivial.

**Justification For Why Not Lower Score:**

N/A

---

### Decision · Program_Chairs · 2024-01-16

Reject